# The relationship between geographic range size and rates of species diversification

Jan Smyčka ●[1] ✉, Anna Toszogyova ●[1] & David Storch ●[1,2]

Range size is a universal characteristic of every biological species, and is often assumed to affect diversification rate. There are strong theoretical arguments that large-ranged species should have higher rates of diversification. On the other hand, the observation that small-ranged species are often phylogenetically clustered might indicate high diversification of small-ranged species. This discrepancy between theory and the data may be caused by the fact that typical methods of data analysis do not account for range size changes during speciation. Here we use a cladogenetic state-dependent diversification model applied to mammals to show that range size changes during speciation are ubiquitous and small-ranged species indeed diversify generally slower, as theoretically expected. However, both range size and diversification are strongly influenced by idiosyncratic and spatially localized events, such as colonization of an archipelago or a mountain system, which often override the general pattern of range size evolution.

Diversification rate is often assumed to be higher in large-ranged species. The idea that larger ranges have a higher probability of being dissected by barriers promoting allopatric speciation was proposed by Charles Darwin[1], and was further developed[2–5] and supported by formal models[6]. Higher speciation rates of large-ranged species are also assumed by many macroecological and macroevolutionary theories[7–11]. For instance, the Neutral Theory and its derivatives assume that every individual or local population has a certain probability to establish daughter species[12–15], and large ranges host more individuals or local populations, thus increasing speciation probability. Moreover, larger species ranges should also lead to lower probability of extinction[2,6,12,16–18], so that net diversification rate (speciation minus extinction) is universally expected to be higher in large-ranged species. All together, these expectations suggest macro-evolutionary source-sink dynamics, where the large-ranged species are driving diversification, and the existence of small-ranged species is maintained by the continuous influx of new species emerging from speciation events that involve range size reduction.

The view that small-ranged species diversify slower, however, seems to be at odds with empirical evidence. Many regions of the world, so called neoendemic hotspots, host evolutionarily young small-ranged species[19–26]. These neoendemic hotspots are often claimed to be the centres of ongoing diversification, and thus areas of particular interest in terms of protecting future cladogenetic evolutionary potential (i.e. the ability of species and lineages to diversify in the future[27]). The link between diversification rate and endemism (defined as local concentration of range-restricted species) is believed to be so universal that the endemism itself has been sometimes used as an indicator of faster diversification in the absence of phylogenetic data (e.g.[28,29]). Besides local endemic hotspots, range size has been shown to be negatively associated with diversification rate metrics in global data[30]. For instance, if we plot the diversification rate metric (DR[31]; an inverse value of species isolation on the phylogenetic tree) against range size across all mammals, we get a significant negative relationship. The negative relationship, although not always significant, holds also for the largest mammalian orders analyzed separately (see Fig. 1, Supplementary Fig. 1 for bootstrap-based envelopes and Supplementary Fig. 2 for alternative analysis using BAMM estimates of diversification).

There are two non-exclusive explanations for this discordance between theory and data. The first one is that the empirical negative correlation between range size and phenomenological estimates of diversification rate (Fig. 1 and e.g.[30,32]) does not result from small-ranged species diversifying faster, but instead from fast diversifying

[1]Center for Theoretical Study, Charles University and the Academy of Sciences of the Czech Republic, CZ-11000 Prague, Czech Republic. [2]Department of Ecology, Faculty of Science, Charles University, CZ-12844 Prague, Czech Republic. ✉e-mail: smyckaj@gmail.com

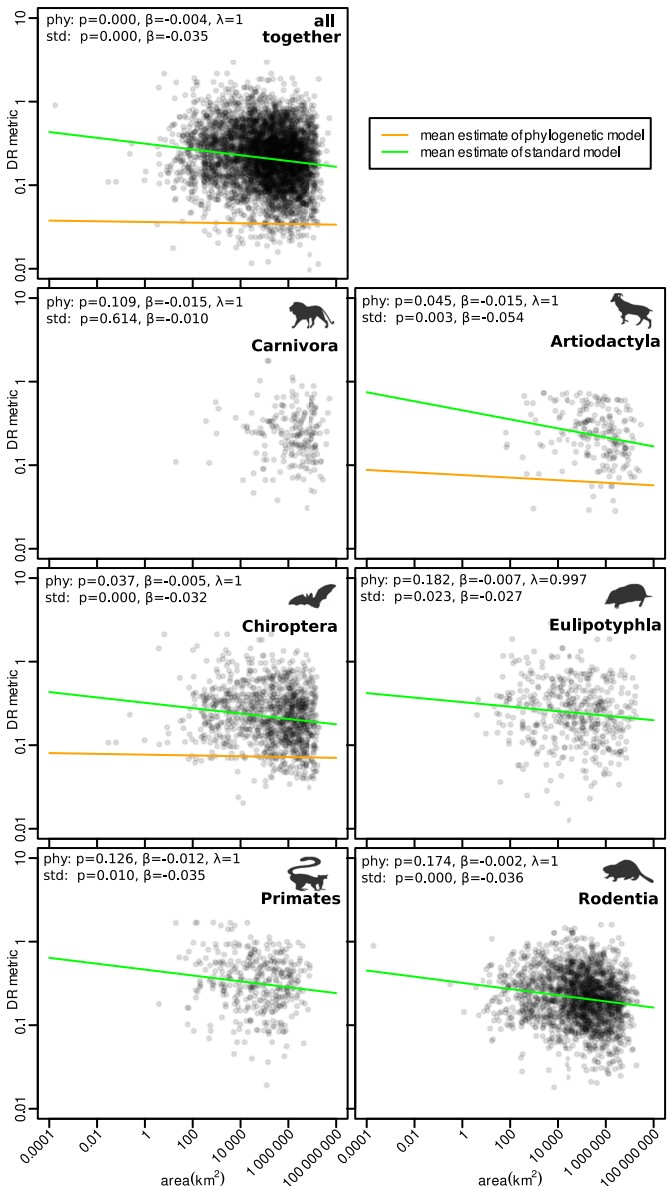

**Fig. 1 | The relationship between range size of individual species and phenomenological estimate of tip diversification rate (DR metric) for all mammals considered together and for large mammalian orders separately.** The statistical significance and the slope parameter estimates are given both for phylogenetic (phy) and standard (std) linear model, and Pagel lambda estimate is provided for the phylogenetic model. Statistical significance provided by both phylogenetic and standard linear model is based on comparing t-statistic of the estimated regression slope against the two-sided Student distribution assuming zero slope. The mean estimate of the slope is negative in all the cases under both standard and phylogenetic linear model. Only the regression lines with slopes different from 0 at $p < 0.05$ are shown. Note that the phylogenetic linear models systematically underestimate the DR values, which is related to the definition of DR metric and its interference with phylogenetic autocorrelation. For details on this behavior, and also bootstrap envelopes around the regression estimates, see Supplementary Fig. 1. The analyses were performed on log-transformed data and both axes have logarithmic scale. The animal contours are adapted from the PhyloPic database (www.phylopic.org). Source data are provided as a Source Data file.

species becoming small-ranged in the process. Common speciation mechanisms, such as vicariant, peripatric, or different types of sympatric speciation, typically lead to the situation when one or both daughter species have considerably smaller population or range size than the mother species. This would mean that the range size evolution cannot be regarded as a continuous process taking place only along the phylogenetic branches (i.e. anagenetic[33–35]), as assumed in classical continuous trait evolution models and respective phylogenetic correction methods[36,37]. Instead, it involves also a shrinkage of the ranges at the point of speciation, i.e. a cladogenetic component, causing many newly emerged species to be small-ranged. This explanation is in line with macroecological and evolutionary theories[7–15] and process-based models of range size evolution[6], which assume that

diversification is driven by large-ranged species. Importantly, it would imply that the small-ranged species with high values of tip diversification rate metrics, such as the ones in the neoendemic hotspots, have emerged from the fast diversification of large-ranged ancestors, and are less likely to diversify in the future, because their ranges have already shrunk[38].

Another explanation is that the theory discussed above and the process based models fail to capture some important features of the real world, and that small-ranged species indeed diversify disproportionately fast under some circumstances. For instance, geographic domains with high density of internal geographic barriers and high level of geographic or ecological isolation, such as archipelagos or mountains, may stimulate faster speciation[32,39–41], often further

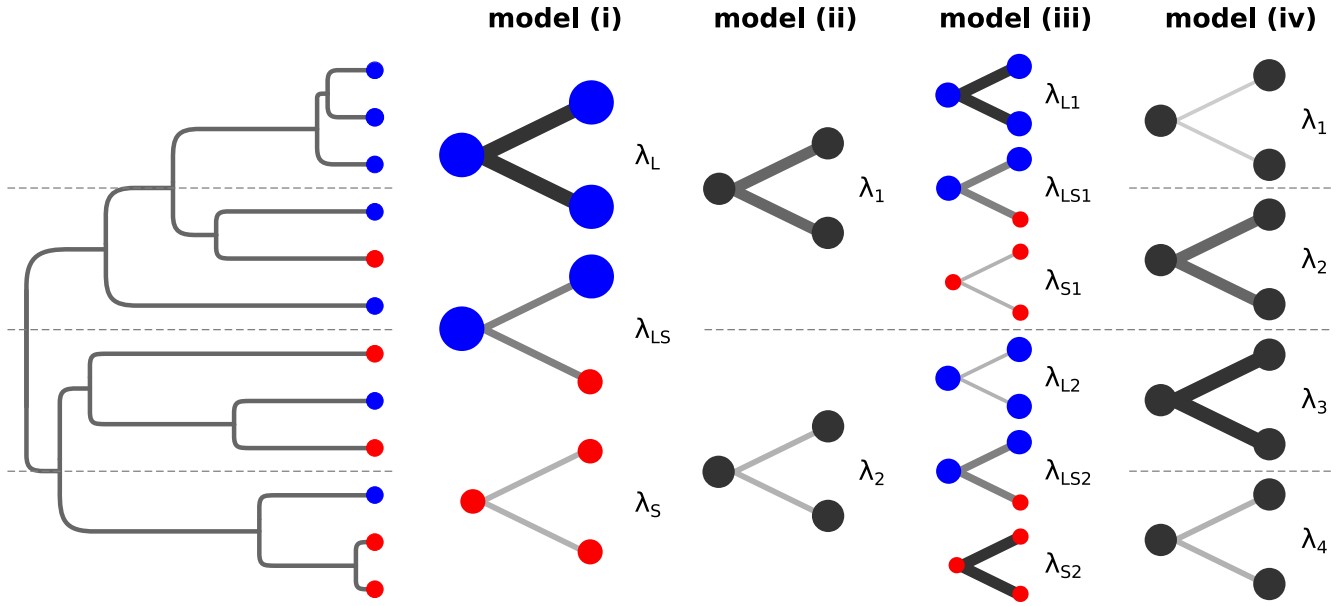

**Fig. 2 | Schematics of the state-dependent diversification models used.** The dynamics of range size evolution were explored using phylogenies with range size categories mapped to their tips (large- vs small-ranged species depicted in blue and red, respectively). We use four speciation-extinction state-dependent models that allow for various diversification dynamics and assume that different parts of the phylogeny may evolve under different concealed regimes controlled by specific parameter sets (separated by the dashed lines). The four models differ in speciation, extinction and state transition structure with the total number of free parameters increasing from model i to iv (see Methods for full description), but the fundamental differences can be explained on the structure of the speciation process: Model (i) assumes different rates for speciation events where a large-ranged mother species produces two large-ranged daughters ($\lambda_L$), for events where a large-ranged mother species gives rise to one large-ranged daughter and one small-ranged daughter ($\lambda_{LS}$), and for events where a small-ranged mother gives rise to two small-ranged daughters ($\lambda_S$; depicted by color bifurcations, with line widths illustrating rate values). This single range size evolution regime is applied across the whole phylogeny. Model (ii) assumes that speciation rate is independent of range size of species involved, but that there are two speciation rates ($\lambda_1$ and $\lambda_2$) in different parts of the phylogeny (separated by the central dashed line matching the one in the phylogeny). Model (iii) assumes that speciation rate depends on range sizes, similarly as in model i, and moreover, there are two such range size evolution regimes in different parts of the phylogeny (separated by the central dashed line matching the one in the phylogeny); e.g. it is possible that speciation of large-ranged species is faster than of small-ranged species in some parts of the phylogeny ($\lambda_{L1} > \lambda_{S1}$), and vice versa in other parts ($\lambda_{L2} < \lambda_{S2}$), as illustrated in the figure. Model (iv) assumes that speciation rate is independent of range size of species involved, and that there are four such rates ($\lambda_1$, $\lambda_2$, $\lambda_3$ and $\lambda_4$) in different parts of the phylogeny (separated by all the dashed lines matching the ones in the phylogeny).

promoted by fluctuations of climate[39,42,43] or sea levels[44,45]. At the same time, maximum range sizes of respective lineages remain limited to the sizes of the domains, and the evolutionary dynamics of range size thus strongly reflect the geographic settings[35]. Such internally structured geographic domains would indeed be the centers of active diversification of small-ranged species, and would deserve the effort towards conserving future evolutionary potential.

Here we explore the relationship between diversification and range size in a global dataset of mammals with respect to both these explanations. For this purpose, we developed a likelihood-based range size-dependent-diversification models accounting for range size changes within and outside the speciation process (i.e. both cladogenetic and anagenetic[46]). Equivalent approaches have been previously used for modeling the evolution of species characteristics with presumably similar dynamics, such as niche breadths[47,48]. Our diversification models suggest that the large-ranged species diversify faster on average when accounted for cladogenetic shrinkage during the speciation process. They also reveal that the previously reported relationship between species range size and the phenomenological metrics of diversification may easily result from analytical artifacts. Further, we perform a series of ancestral state reconstructions and explore the residual variability of diversification rates in our models (using hidden or concealed states sensu[49,50]) to identify the exceptions from the general pattern. The identified exceptional small-ranged radiations indeed often take place in complex geographical domains as are the oceanic archipelagos or mountains.

## Results
### State-dependent diversification models

We explored diversification and range size dynamics in mammals using four diversification models with or without the dependency of diversification rates on range size, and with different numbers of concealed diversification regimes across the phylogeny (sensu[50], see Fig. 2 and Methods for the description of the four models). We fitted each of the models to the phylogeny of terrestrial mammals[51], and also separately to mammalian orders with more than 200 extant species to explore order-specific patterns. As our diversification models assume discrete range size categories, we categorized species to large-ranged and small-ranged based on IUCN range size data, using the threshold 177,907 km² (approximately the size of Sulawesi island), which represents the median species range size in our dataset. We also tested for alternative thresholds of 250,000 km² and 50,000 km², and for the medians specific to individual orders, yielding similar results (see Fig. 3 and Supplementary Fig. 3). In addition, we evaluated whether our results are robust to the addition of up to 30% of cryptic species into our dataset (see Methods and Supplementary Fig. 4 for details). The patterns of range size across the phylogeny of all mammals together were best explained by the model with two range size evolution regimes (model iii as in Fig. 2, Akaike weight 1.00). This was the case also for large mammalian orders analyzed separately (see Table 1), with the exception of Artiodactyla, where the range size evolution model with a single regime (model i) and the null model with two diversification regimes independent of range size (model ii) received marginally better support. The high support for model iii suggests that range

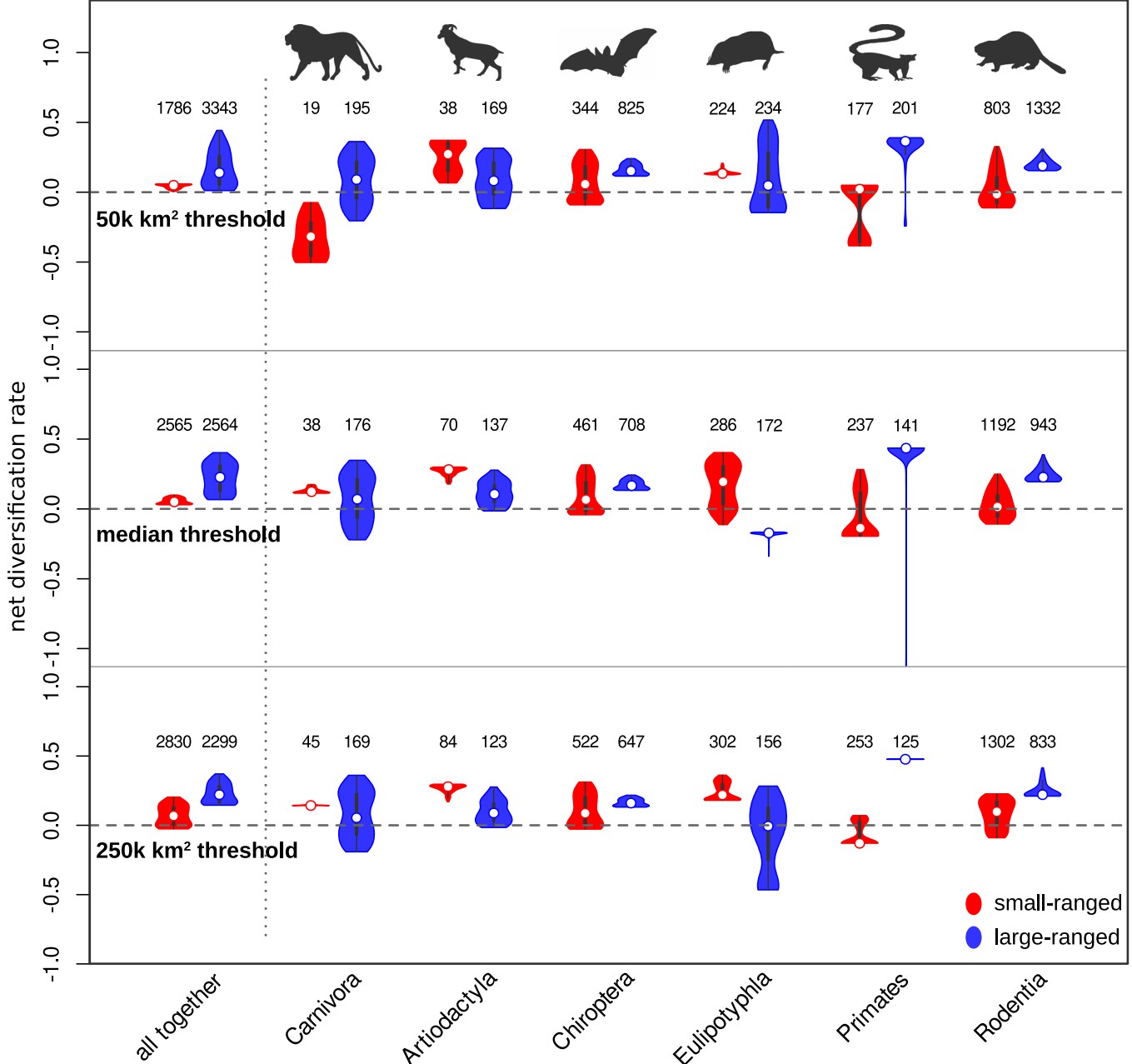

**Fig. 3 | Tip estimates of net diversification rate (rate of speciation minus rate of extinction) for individual species from a cladogenetic range size-dependent diversification model with concealed states (model iii).** The three panels represent estimates using 50,000 km², 177,907 km² (median) and 250,000 km² thresholds defining small- and large-ranged species. Higher net diversification rate values reflect faster-diversifying species, overlap between red and blue violins suggests that the depicted taxon contains large- and small-ranged species diversifying at similar rates, and negative values indicate species that are evolutionary sinks. The dots in the violin plots represent medians, the boxes represent interquantile range, the whiskers represent 1.5x interquantile range, and the smooth curves represent kernel density estimates. The numbers of large- and small-ranged species in each taxon, based on which the violin plots were constructed, are depicted above the violins. The animal contours are adapted from the PhyloPic database (www.phylopic.org). Source data are provided as a Source Data file.

**Table 1 | Akaike weights for different variants of cladogenetic range size-dependent diversification models, for all mammals together, and for large mammalian orders separately. The highest supports and supports with ΔAIC < 2 are depicted in bold**

| Model | all mammals | Carnivora | Artiodactyla | Chiroptera | Eulipotyphla | Primates | Rodentia |
|---|---|---|---|---|---|---|---|
| (i) range size-dependent | 0.00 | 0.00 | **0.36** | 0.00 | 0.00 | 0.00 | 0.00 |
| (ii) null with two concealed regimes | 0.00 | 0.13 | **0.44** | 0.00 | 0.00 | 0.00 | 0.00 |
| (iii) range size-dependent and two concealed regimes | **1.00** | **0.87** | 0.20 | **1.00** | **1.00** | **1.00** | **1.00** |
| (iv) null with four concealed regimes | 0.00 | 0.00 | 0.00 | 0.00 | 0.00 | 0.00 | 0.00 |

size generally affects diversification rates, but its effect varies across different parts of the phylogeny, which evolve under different range size-dependent diversification regimes.

We further explored the variation in range size-dependent diversification rates by analyzing parameter estimates and predictions of model iii, and their differences across mammalian orders. In all mammals taken together, both diversification regimes consisted of relatively fast diversifying large-ranged species and relatively slowly diversifying small-ranged species. This pattern was not largely influenced by the definition of large/small range threshold (Fig. 3). An important portion of the diversification of large-ranged species was due to speciation events that resulted in one large- and one small-ranged daughter species ($\lambda_{LS}$ *sensu* Fig. 2). As indicated by the estimates of model parameters (Supplementary Fig. 5, Supplementary Data 1), the per-lineage rate of speciation where a large-ranged species produced one large- and one small-ranged daughter ($\lambda_{LS}$) was between 0.04 and 0.40 events/Ma, whereas the rate of speciations where large-ranged species produced two large-ranged daughters ($\lambda_L$) was between 0.00 and 0.05 events/Ma, and the rate of speciations where a small-ranged species split to two small-ranged daughters ($\lambda_S$) was between 0.1 and 0.2 events/Ma.

### Simulation experiment

To test whether the macroevolutionary dynamics represented by these parameter values are likely to generate a negative relationship between DR and range size, similar to one reported in Fig. 1, we performed a simulation experiment (see Methods for details). All 100 phylogenies simulated in this experiment showed a negative relationship between the DR metric and range size, although with slightly different regression slopes than the real data. This relationship was significant in all cases using a standard linear model, and also in most cases using phylogenetically corrected regression (Fig. 4).

### Tip estimates of diversification and exceptional lineages

For large mammalian orders analyzed separately, there are notable deviations from the general dynamics detected in all mammals analyzed together. Multiple clades within Artiodactyla and Chiroptera reveal faster diversification in small-ranged species than in the large-ranged species, and almost all small-ranged Eulipotyphla diversify faster than the large-ranged species (see the median values in Fig. 3). For examples of clades consisting of small-ranged species with highest tip estimates of diversification rates in Artiodactyla and Chiroptera see Fig. 5, for a complete list of species and estimates of their tip diversification rates, see Supplementary Data 2. These order-specific patterns are generally robust to the redefinition of large/small threshold to 250,000 km² or 50,000 km² (Fig. 3). Moreover, they mostly hold even if the analyses are repeated with order-specific medians, thus controlling for the differences in range sizes and dispersal capacities of different orders (Supplementary Fig. 3). An important exception is the order Eulipotyphla that gives inconsistent results when using the alternative thresholds, with large-ranged species revealing either negative diversification rates with low variation (median thresholds in Fig. 3) or a wider variety of diversification rates (50,000 km² and 250,000 km² thresholds in Fig. 3 and order-specific median in Supplementary Fig. 3). However, under all these alternative thresholds, an important portion of large-ranged species in Eulipotyphla is always estimated to have lower diversification rates than the small-ranged species.

Besides Eulipotyphla, very low diversification rates are predicted for multiple large-ranged species in Carnivora and one species of Primates (Fig. 3). In some cases, the net diversification rate in these large-ranged species is estimated to be negative, meaning that such species form an evolutionary sink and are more likely to go extinct than further speciate. The examples of large-ranged species with the lowest predicted diversification rate from Eulipotyphla, Carnivora and Primates

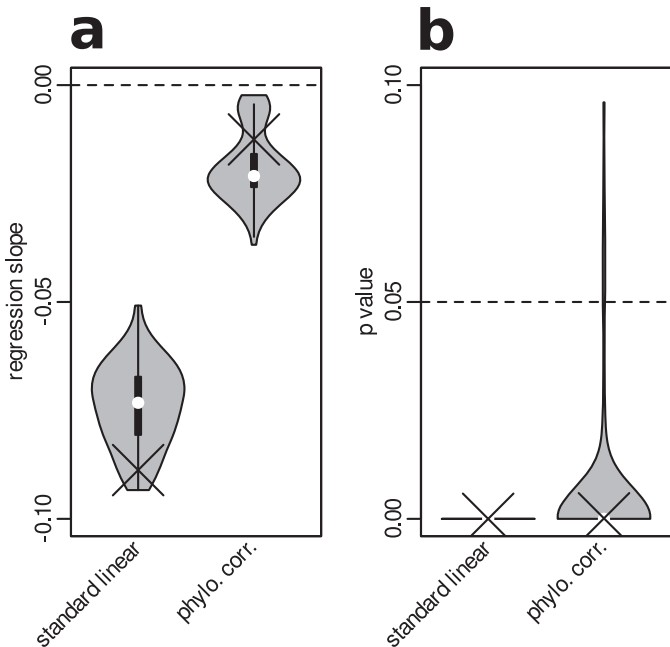

**Fig. 4 | The results of the simulation experiment using 100 phylogenies generated with model iii parameters.** Panel (**a**) shows the slopes of standard and a phylogenetically corrected linear model of the relationship between DR metric and range size category, similar to Fig. 1. Panel (**b**) compares *p* values of the respective models. The crosses represent slopes and p-values retrieved from the real mammalian phylogeny. The violins depict distribution of slopes and p-values retrieved from 100 phylogenies independently simulated based on model iii parameters. The dots in the violin plots represent medians, the boxes represent interquantile range, the whiskers represent 1.5x interquantile range, and the smooth curves represent kernel density estimates. Statistical significance provided by both phylogenetic and standard linear model is based on comparing t-statistic of the estimated regression slope against the two-sided Student distribution assuming zero slope. Source data are provided as a Source Data file.

are *Uropsilus gracilis* (shrew mole from SE China), *Nandinia binotata* (palm civet widely distributed in subsaharan Africa) and *Tarsius bancanus* (tarsier widespread across Greater Sunda islands). The exhaustive list of all species ordered by their estimated diversification rates can be found in Supplementary Data 2.

## Discussion

We have explored the evolutionary dynamics of range size in mammals using state-dependent diversification models. Unlike alternative approaches[31,52–54], our models explicitly account for range size changes during speciation, a phenomenon expected under common types of speciation, such as vicariant and peripatric[6] speciation, and also for most sympatric speciation mechanisms. The parameter estimates from our model for all mammals suggest that large-ranged species diversify on average faster than small-ranged species, and that this large-range diversification is indeed strongly driven by speciation events linked with range size change, when a large-ranged mother species produces one large- and one small-ranged daughter. More specifically, the rate of these speciations ($\lambda_{LS}$ *sensu* Fig. 2) is comparable to the rate of speciations when a small-ranged species produces two small-ranged daughters ($\lambda_S$), and higher than speciations when large-ranged mother produces two large-ranged daughters ($\lambda_L$), so that the total speciation rate of large-ranged species ($\lambda_{LS} + \lambda_L$) is typically higher than the total speciation rate of small-ranged species ($\lambda_S$).

Importantly, these evolutionary dynamics with large-ranged species diversifying faster due to speciations involving range size change can produce a significant negative relationship between DR metric and range size, as demonstrated with our simulation experiment (Fig. 4).

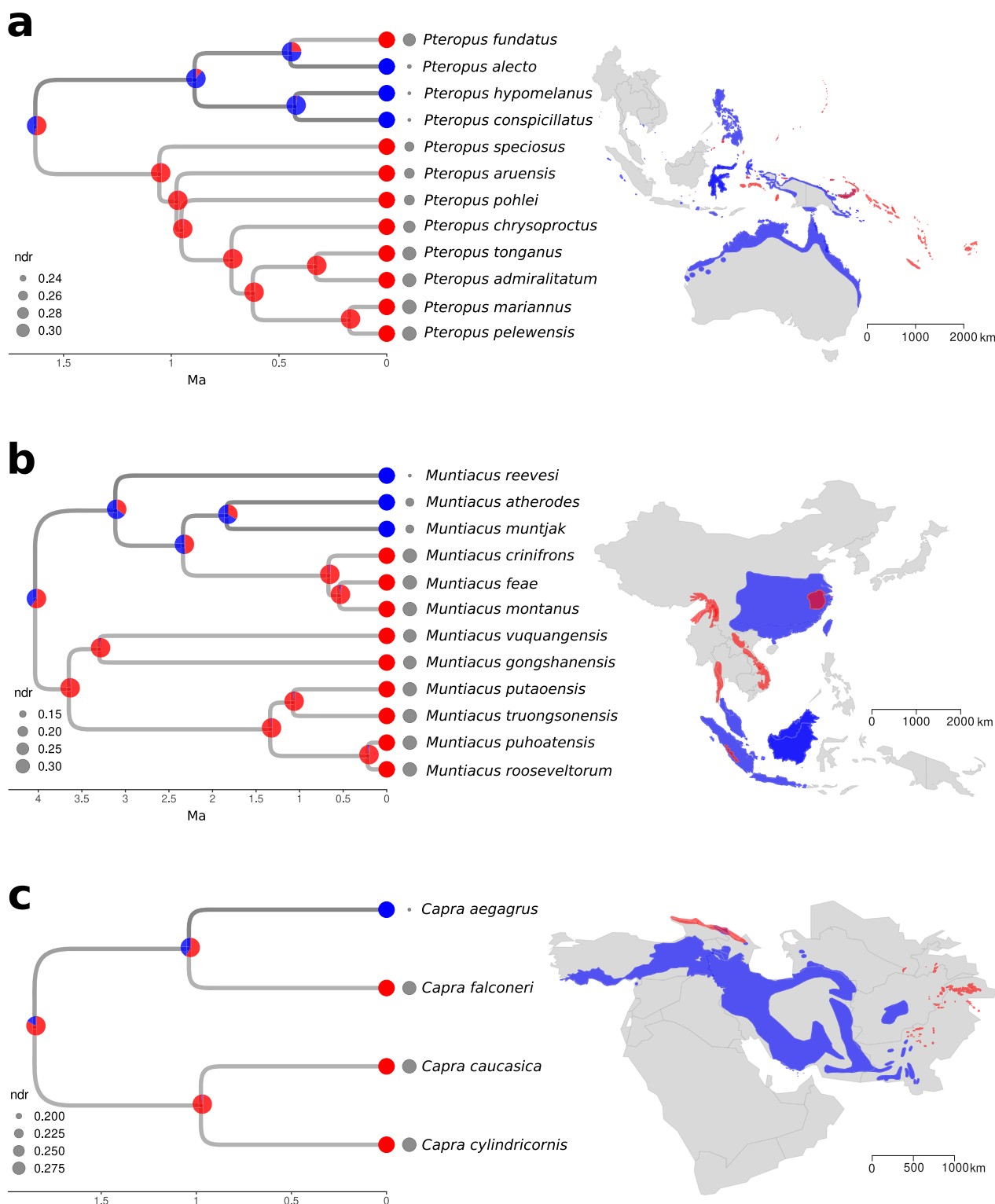

**Fig. 5 | Examples of the fastest radiations of small-ranged species.** The panel (**a**) shows a phylogeny of fast radiation in flying foxes (*Pteropus*), the panel (b) fast radiation in muntjacs (*Muntiacus*) and the panel (c) a fast radiation in goats (*Capra*), along with geographic ranges of individual species. Small-ranged species are depicted in red, large-ranged species in blue. The pie plots at the nodes represent the ancestral state probabilities based on model iii estimates. Sizes of the grey dots represent the estimates of net diversification rate (rate of speciation minus rate of extinction) for individual species.

This provides an explanation for the counter-intuitive pattern that small-ranged species have higher values of DR metric than large-ranged species (Fig. 1), although they are expected to diversify slower[2,6,10–12,15]. Although we mostly explored the behavior of DR metric in this paper, it can be expected that similar considerations apply also to other phenomenological metrics of diversification (see results for BAMM estimates in Supplementary Fig. 2). The high phenomenological diversification metrics correctly describe that many small-ranged species emerged recently. However, they are unable to reflect that many young small-ranged species emerged from large-ranged mothers that indeed have higher diversification rates, as theoretically expected.

The prevalence of range size changes during speciation may also explain why the results of studies of range size heritability are mixed and often controversial (e.g. refs. 33–35,55). The commonly used models of continuous trait heritability (Brownian, Ornstein-Uhlenbeck and similar) assume only anagenetic changes along the branches, not at the point of speciation. We overcame this constraint by discretizing range size into the categories of large and small ranges. This allowed us to use the discrete models of trait evolution that can account for both anagenetic changes and the changes at the point of speciation. It would be theoretically possible to use more than two range size categories in the cladogenetic diversification models, mimicking the continuous range size variation. However, such models would impose significant computational and conceptual challenges (see Methods for details), which is why we prioritized the two-category model. An important limitation of our two-category model is that some speciation events with an actual range size change are not reflected by the changes of range size category, because none of the daughter species passes the large/small threshold. According to the simulations (Supplementary Fig. 6), only about 20% of vicariant speciation events lead to one of the daughter species changing the range size category, although this percentage is expected to be higher in peripatric speciation events. Importantly, even the subset of range size change speciation events identified in the two-category model ($\lambda_{LS}$) has the rates on the same orders of magnitude as the anagenetic transitions between the range size categories ($q_{LS}$ and $q_{SL}$). It suggests that range size changes at speciation may be a key component of range size evolution, and future studies of continuous range size heritability should either control, or explicitly account for this process.

Although the large-ranged species diversify on average faster than small-ranged species for all mammals considered together, the relationship between range size and diversification rate is quite variable and actually inverted in many mammalian clades, including whole orders (Fig. 3). The differences between mammalian orders do not seem to follow any obvious pattern, such as a systematic difference between orders with better (Carnivora, Artiodactyla, Chiroptera) and worse (Eulipotyphla, Primates, Rodentia) dispersal capacity or between orders differing in their median range sizes (see Supplementary Fig. 3). Similarly, there do not seem to be systematic differences between mostly tropical (Chiroptera, Primates) versus mostly temperate mammalian orders, suggesting that our results are not driven by a combination of the Rappoport rule[56,57] and an inverted latitudinal gradient of diversification[58–61]. Moreover, the latitudinal gradient of range sizes in our dataset is weak (Supplementary Fig. 7) compared to the gradient of latitudinal extents[56], and the evidence of a latitudinal gradient of diversification in mammals is mixed and dependent on used methods and their assumptions (increasing in[61] vs. decreasing in[62]).

For an easier interpretation, we can divide the deviations from the overall mammalian trend into two cases. The first case comprises small-ranged species and lineages that diversify exceptionally fast, often leading to phylogenetically localized radiations. In bats (Chiroptera), for instance, flying foxes (*Pteropus*; Fig. 5a) or *Miniopterus* bats radiated on islands of the Indian and Pacific Ocean. These radiations might have been stimulated by the colonization of the oceanic archipelagos, triggering the speciation jumps from one island to another (as discussed for *Pteropus*[63]), and at the same time limiting the range size of resulting species to individual islands. This mechanism is likely linked with bat ability to fly, as bats are the only mammalian group that was shown to commonly diversify in insular systems[64], similar to birds which also reveal faster diversification on islands compared to mainlands[32,40]. It is not clear whether colonization of an insular system is a sufficient condition of small-ranged radiations in bats, or just one of the preconditions, but the existence of small-ranged slowly diversifying island species and lineages (e.g. *Emballonura serii*) would advocate for the latter.

The radiations of small-ranged bat species may also take place in island-like terrestrial systems, such as topographically complex landscapes of Eastern Africa (e.g. bat genus *Scotophilus*). Terrestrial habitats with a pronounced geographic structure might have stimulated diversification of small-ranged species and lineages also in flightless orders like ungulates (Artiodactyla). As identified by our analysis, this likely happened in muntjacs (*Muntiacus*) in complex karstic landscapes of south-eastern Asia (Fig. 5b) or in goats (*Capra*) inhabiting high mountains of Eurasia (Fig. 5c). A specific case is the island radiation of pigs (*Sus*) in the archipelago of Philippines, but recent phylogeographic evidence suggests that the dispersal of pigs in the Philippines was driven by humans[65]. Fast-radiating and small-ranged bats and ungulates are associated with geographic domains that contain barriers, such as archipelagos and mountains, which stimulate speciation and at the same time prevent species from expanding their ranges. Such spatial concentration of barriers constitutes a clear deviation from theoretical models of range or population formations[6,12], that typically assume random placement of reproductive barriers. It would be interesting to explore which intensity of internal and external barriers (e.g. spacing of islands with an archipelago and isolation from mainland) is necessary to trigger radiations of small-ranged species. Unfortunately, the anecdotic character of insular and other radiations in our dataset makes a more formalized analysis of this phenomenon difficult.

The other deviation from the overall trend of fast diversification of large-ranged species comprises large-ranged species that diversify exceptionally slowly. At the extreme, such species may have negative net diversification rate and thus represent evolutionary sinks (they are more likely to go extinct than speciate). These species do not appear to share obvious biogeographic features, as they range from temperate species presumably influenced by post-glacial climate dynamics (*Uropsilus gracilis*), to widespread tropical species with ranges covering entire biomes (*Nandinia binotata*), or species inhabiting multiple islands (*Tarsius bancanus*). Some of these species might have only recently expanded, after they have spent the majority of their evolutionary history as range-restricted[66]. Other species might have developed population-level mechanisms allowing efficient genetic mixing across the large geographic range, such as intensive long distance migration[67]. In addition to this, up to 30% of mammalian species may be cryptic[68], which might be the case especially for large-ranged species complexes in Chiroptera, Eulipotyphla and Rodentia. Our results are robust to an addition of up to 30% of cryptic species in our dataset (Supplementary Fig. 4, see Methods for details), but the Linnean shortfall (*sensu*[69]) may still influence the above-reported idiosyncratic exceptions. The prevalence of cryptic species within the large-ranged mammalian taxa, and also particular mechanisms of the maintenance of large ranges over long timescales could be explored using quantitative molecular phylogeographic data, once these get available for a significant part of global mammalian biota.

The theoretically expected positive relationship between range size and diversification rate may thus get locally overridden due to idiosyncratic geographical settings (see also[32]). The list of above-

reported exceptions is not exhaustive and we expect that considering different phylogenetic (*sensu*[70]) or spatial scales (the threshold defining small and large ranges, e.g. the situation shown here for Eulipotyphla), may highlight other idiosyncrasies, potentially reverting the local or even global patterns. However, the mechanisms causing the positive relationship between range size and diversification as proposed by the ecological theory[2,6,15] are likely to take place even in the cases that we identified as the exceptions, and they only got obscured by the local settings. For instance, we can assume that the artificial reduction of the range of individual species by anthropogenic impact would decrease rather than increase its diversification potential (as discussed in[11]) even in the lineages where small-ranged species diversify faster. The reason is that the link between small ranges and fast diversification in these lineages is likely not directly causal, but is mediated by the coincidence of small ranges and geographic barriers on one hand and fast diversification and geographic barriers on the other hand. Although they do not reject the existence of general mechanisms, the idiosyncratic deviations described above have important implications for the comparison of outcomes of theoretical models such as neutral theory[15] or models describing evolutionary range dynamics[6] with real-world data. Such comparisons should account for geographic features such as islands or mountains, either explicitly (as is e.g.[14]) or by selection of the scales and study systems where such effects are minimized.

Our findings have important consequences for estimating cladogenetic evolutionary potential (i.e. the ability of further diversification) of lineages or species assemblages at particular regions based on range sizes and phenomenological diversification metrics[19–23,27,29]. We have shown that large-ranged mammals diversify on average faster. But this relationship is fairly variable and depends on phylogenetic and biogeographic context, suggesting that range size itself is not an efficient predictor of cladogenetic potential. At the same time, our results suggest that the assemblages of species with short phylogenetic branches may not necessarily indicate an ongoing in situ radiation in the focal area, but also historical range-splitting that has already exhausted[38], or common peripatric speciation from the adjacent source area (e.g.[71]). Each of these processes have different implications for conservation planning, but they cannot be easily distinguished without additional information. The phenomenological diversification measures proved to be unreliable as predictors of diversification potential also in crossvalidation studies using fossil phylogenies[72]. Our results suggest that one possible way to improve such predictions is integrating the range size and phylogenetic data using more realistic assumptions about range size evolution (e.g. cladogenetic changes of range size), and also accounting for local spatial and phylogenetic context. A potentially powerful approach for disentangling incipient from exhausted radiations would comprise integrating phylogenetic and range data with the information on genetic structure of individual species. Past efforts in this direction were often limited by the resolution of population genetic data[21,73] or by spatial and phylogenetic extent they covered[74]. However, the fast development of genomics may enable identifying global centers of incipient diversification in the future, using population-level genetic data.

In conclusion, we have shown that although mammal diversification rate is, in accord with the theory, generally higher in large-ranged species, the relationship between range size and diversification rate is variable and likely depends on particular geographic settings. Idiosyncratic geographic settings, such as insularity, likely promote diversification rate on one hand, but limit maximum range size of species on the other hand. Range size itself commonly changes during the speciation process, and consequently the statistical relationship between range size and measures of diversification rate based on branch lengths is not informative in terms of the factors that affect diversification. Moreover, the variable relationship between range size and diversification complicates the efforts for estimating the cladogenetic evolutionary potential of species and lineages based on range size and phylogenetic relationships. Our results thus emphasize that process-based understanding of range size dynamics and diversification, and also explicit consideration of geographic space, are crucial for both macroevolutionary theory and large-scale conservation planning.

## Methods

### Data

We extracted the range size information of all terrestrial mammals from the IUCN geographic distribution database[75]. In this process, we united all polygons depicting individual populations by species, excluding populations tagged as (possibly) extinct, (re)introduced, vagrant, non-breeding, presence uncertain or origin uncertain. We calculated the range size of each species and mapped these species-level data on the tips of a maximum credibility tree of mammals from Upham et al. (2019)[51]. The data processing was performed with R 3.6.3, using packages dplyr 1.0.7[76], rgdal 1.5-23[77], raster 3.4–13[78], maptools 1.1-2[79], cleangeo 0.2–4[80], ape 5.6-1[81] and phytools 0.7–80[82].

Despite common usage in macroecological and macroevolutionary studies (e.g.[30,83]), our mammalian dataset might be prone to both geographic biases (i.e. Wallacean shortfall *sensu*[69]), and issues related to taxonomy and definition of species (i.e. Linnean shortfall *sensu*[69]). The Wallacean shortfall is likely not limiting for our inference, due to the categorization of range sizes into the above and below median categories in the downstream analyses. It is difficult to imagine that the imperfection of IUCN range data would lead to common misplacement of species above or below the median. Moreover, our sensitivity analyses show that the presented results are robust across the relatively large extent of small/large threshold definitions (see Fig. 3 and Supplementary Fig. 3).

The Linnean shortfall may impose a more important limitation. It is possible that some species used in the IUCN dataset or Upham et al. (2019)[51] phylogeny are in fact complexes of cryptic species. A recent study suggests that up to 30% of mammal species may actually be cryptic[68]. We explored the sensitivity of our diversification inference to the presence of cryptic species by rerunning the downstream diversification analyses on the datasets with 10% or 30% of large-ranged species artificially split into the pairs of cryptic sister species. These analyses suggest that the general results of diversification analyses are robust to the possible occurrence of cryptic species (Supplementary Fig. 4). However, we cannot exclude that the outlier species or exceptional groups we discuss throughout the paper may in some cases result also from imperfect taxonomical treatment.

### Phenomenological diversification rate metric regression

We calculated the diversification rate metric (DR metric[31]) of each species in our dataset as an inverse value of equal splits evolutionary distinctiveness metric[84], as implemented in R package picante 1.8.2[85]. The DR metric thus reflects the evolutionary isolation of species from all other species of the phylogeny, with the most isolated species having the lowest values. As an alternative, we also calculated net diversification estimates for the tips using BAMM[52,86]. We ran the BAMM analysis for 100 M iterations, with 10k thinning and 50% burn-in, otherwise mimicking the analysis setup from Upham et al. (2019)[51]. We checked that the effective sample size of the resulting chain was well above 500 for all the monitored parameters, and also assessed the convergence visually.

The phenomenological diversification estimates of every species (based on DR or BAMM) were fitted against log-transformed range size values using standard linear model, but also with the generalized least

square model with range size as a predictor and error structure reflecting Brownian evolution weighted by Pagel lambda parameter (*sensu*[36]), using R package phylolm 2.6.4[37]. The robustness of estimates from both standard and phylogenetic models was evaluated using 100 bootstrap replicates (Supplementary Fig. 1). The calculations were run on all mammals together, and then separately for the mammalian orders containing more than 200 species (6 orders).

## State-dependent diversification models

We used a series of state-dependent speciation-extinction models (SSE[87]) to determine whether the range sizes are linked with specific diversification rates, and to estimate the parameters of this linkage. Our SSE models consider a phylogeny with mapped tip states as a realization of a branching process, where each lineage can split into two lineages maintaining mother state (constant-state speciation), split into two lineages with the change of states (state-change speciation), disappear (extinction) or transit from one state to another (anagenetic state change). An advantage of SSE methodology over approaches based on hypothesis-free diversification measures such as DR[31] or BAMM[86] estimates is that it can explicitly accommodate state-change speciation events, e.g. vicariance or peripatric speciations where the changed range sizes of daughter species emerge as a direct consequence of speciation event. The inclusion of range changes during speciation in our models, however, brings one compromise – the range size cannot be efficiently modeled as a continuous trait, but needs to be discretized. Here we used categorization of range sizes to large and small using the median (177,907 km$^2$; approximately the size of Sulawesi) as the splitting point, a strategy previously used in a similarly designed study of the relationship between diversification and niche breadths[47]. To test the robustness of the results to this choice, we reran our analyses with alternative thresholds of 250,000 km$^2$ and 50,000 km$^2$. We also explored whether the order-specific results change when we use range size medians for individual orders instead of the common threshold for all mammals (see Supplementary Fig. 3). This allowed us to test whether the order-specific differences are driven by the differences in typical range size or dispersal capacity of individual orders, or by other factors.

To test the linkage between range size and diversification rates, we compared four SSE models (see Fig. 2) differing by the amount of total diversification rate variability, and by the amount of diversification rate variability linked with range size:

Model (i) assumes that the events where a large-ranged species speciates to two large-ranged daughters ($\lambda_L$), and events where a small-ranged species speciates to two small-ranged daughters ($\lambda_S$) have different evolutionary rates. This model also accounts for speciation events where a large-ranged species gives rise to one large-ranged and one small-ranged daughter ($\lambda_{LS}$), having its own rate, and representing speciations with range size change. The speciation events where a large-ranged species gives rise to two small-ranged species are not represented in this model, since a preliminary analysis suggested that the parameter characterizing speciation of large- to two small-ranged species would be hardly identifiable. More importantly, the speciation events where both parts of the split large range would simultaneously pass below the median threshold are expected to be relatively rare. The simulated splitting of ranges in our dataset suggests that such events would form less than 10% of all vicariant speciations (Supplementary Fig. 6), and are virtually impossible under the other speciation mechanisms. Extinction rates in this model are also considered different for large-ranged ($\mu_L$) and small-ranged species ($\mu_S$). Apart from diversification process, large-ranged species may shrink to small-ranged along the branches without a speciation event ($q_{LS}$) and vice versa ($q_{SL}$), which is also controlled by two free rate parameters. The model i thus assumes that diversification rates

depend on range size, either positively or negatively. All together, model i has 5 free parameters.

Model (ii) assumes that two different speciation rates take place in the phylogeny and these rates are independent of range size category ($\lambda_1$ and $\lambda_2$). More specifically, it assumes that these rates are linked to two concealed states that define separate diversification regimes, but these states are not associated with range size and cannot be observed on phylogeny tips (*sensu*[50]). Two extinction rates are also linked to these concealed states ($\mu_1$ and $\mu_2$) and are thus independent of range size, and this model does not account for state-change speciation events. Transitions between small and large-range states along the branches are controlled by two rate parameters ($q_{LS}$ and $q_{SL}$), and the same is true also for the transitions between the two concealed states ($q_{12}$ and $q_{21}$). The model ii serves as a null model for model i, taking into account that evolutionary rates may be variable across the phylogeny, but assuming that this variability is not linked to range size. Model ii has 6 free parameters in total.

Model (iii) is a combination of model i and model ii. Like model ii, it accounts for two concealed states of species and lineages that have different diversification regimes. These two diversification regimes, however, do not consist of single speciation and single extinction rate independent of range size as in model ii, but instead they acknowledge the role of range size in the same way as model i. For example, a large-ranged species in concealed state 1 speciates by giving rise to two large-ranged species ($\lambda_{L1}$), one small- and one large-ranged species ($\lambda_{LS1}$), goes extinct ($\mu_{L1}$) or shrinks ($q_{LS1}$) at certain rates, and these rates are different for a large-ranged species in concealed state 2. The transitions between two concealed states are controlled by a pair of rate parameters ($q_{12}$ and $q_{21}$), similarly as in model ii. The model iii thus assumes that diversification rates depend on range size, but that there is a variability in this relationship, represented by different concealed regimes. Taken together, model iii has 12 free parameters.

Model (iv) is similar to model ii, but it assumes four concealed states, and thus four speciation rates, four extinction rates, 12 transition rates between all combinations of concealed states, and rates of range shrinkage and expansion. The model iv is a null model for model iii, assuming four different diversification rates along the phylogeny, none of them being linked to range size. This model has 20 free parameters.

In theory, it would also be possible to define SSE models with more that two range size categories, providing a better approximation of the continuous range size variation. However, there are two reasons why we have avoided this direction, one computational and one conceptual. Introducing e.g. four instead of two range size categories would mean an increase of the number of parameters. This increase would be more than twofold because some of the parameters are related to the transitions among all possible combinations of states. Moreover, using more that two range size categories would also mean using higher number of concealed states. This increase of complexity would make our model parameters likely unidentifiable for the smaller order-specific phylogenies presented in the manuscript (at least for Carnivora and Artiodactyla). At the same time, the resulting models would be extremely demanding on computation time – the parameter optimization takes several weeks to one month of single-core computation time even for the models with two range size categories.

The more important reason for not introducing larger number of range size categories is conceptual. Our model is based on the idea that some speciation events involving range size change happen in mother species with below-threshold range (so they are covered by $\lambda_S$ speciation parameters), some happen in mother species with above-threshold range and both daughter species stay above threshold ($\lambda_L$), and only a subset of speciation events involves a daughter of above-threshold mother crossing below the threshold ($\lambda_{LS}$). It is also possible

that some vicariant speciation events would make both daughter species of above-threshold mother crossing below the threshold simultaneously, but such process seems to be hardly identifiable from the data and our simulations suggest it is relatively rare given the current distribution of mammalian range sizes (Supplementary Fig. 6). However, the situation would be much more complicated if we introduced more than two range size categories. In such a case it would be necessary to tackle how often are the speciation events peripatric, with one daughter maintaining the mother range size category and other daughter getting to any of the multiple small-range categories; how often they are vicariant speciations where both daughters drop in range size, and what is the distribution of the division ratio; and also what happens under different sympatric scenarios. In other words, it would be necessary to explicitly address the frequencies and exact mechanisms of different types of speciations. Despite recent research on this topic[6,88,89], the frequencies of different speciation mechanisms cannot be a priori anticipated and introduced in the modeling process. Such frequencies could in theory also be estimated as free parameters of the models, but this would further increase the computational complexity and demands on the sizes of phylogenies used.

The likelihood functions of the above-described models were defined using the SecSSE 2.0.0 package in R[50]. The advantage of SecSSE over similar approaches for defining and fitting SSE models (e.g. HiSSE[49], castor[90] or diversitree[87]) is that it allows for full user control over the definition of the likelihood function including state-change speciations, and at the same time it can accommodate concealed states. We estimated the maximum likelihood parameter values for each model using the subplex routine[91], which proved to be more computationally demanding, but gives more stable and accurate results than the default simplex optimizer in SecSSE. We compared the four models using Akaike information criterion (AIC) and Akaike weights depicting relative likelihood of each model given the data[92]. This procedure was first carried out for all mammals together, and then separately for all mammalian orders with more than 200 species.

## Tip estimates of diversification and ancestral state reconstructions

We used the parameter estimates from model iii to calculate diversification rates at the tips of the phylogenies that, unlike DR or BAMM, account for range size changes at speciations. To do this, we first calculated net diversification rates in different concealed states for large-ranged species as $\lambda_{Lj} + \lambda_{LSj} - \mu_{Lj}$, and for small-ranged species as $\lambda_{Sj} - \mu_{Sj}$, where j is the concealed state 1 or 2. Further, we estimated probabilities of each tip of a phylogeny being in the concealed state 1 or 2. These probabilities were calculated as relative likelihoods of parameter values when the focal tip is in concealed state 1 or 2. The diversification rate of each tip species was calculated as an average of diversification rates under both concealed states weighted by their probabilities. For Primates, we modified the default relative tolerance of the ODE solver from $10^{-12}$ to $5*10^{-12}$ in the likelihood estimation procedure due to the issues with the convergence of the calculation. Our calculations were performed with the custom code based on package SecSSE (see Supplementary Software 1 for the commented code), but they followed the logic of tip rate calculation implemented and discussed for the package HiSSE[49]. For the diversification rate estimates of each tip, see Supplementary Data 2, for the state probabilities for each tip, see Supplementary Data 3.

We used the estimated tip diversification rates for ordering the species and identifying the presented examples of exceptional lineages (e.g. Fig. 5). In particular, we used monophyletic lineages of small-ranged species with highest diversification rates as examples of fast radiations of small-ranged species in Chiroptera and Artiodactyla. The named examples of slow diversifying large-ranged lineages in Carnivora, Eulipotyphla and Primates were the lineages with the lowest estimated diversification rate in these orders. Due to their nature, these slow diversifying lineages typically consisted of a single evolutionary isolated species. The model iii parameters were also used for estimating range sizes of the ancestral nodes depicted in Fig. 5. The probabilities of each node being large- or small-ranged were calculated as relative likelihoods of the parameter values, when the focal node was in large- or small-ranged state. The node state calculations were performed in R package SecSSE.

## Simulation experiment

We explored whether the parameter values retrieved by SSE models describe the macroevolutionary dynamics that systematically leads to phylogenies with a negative relationship between DR metric and range size, similar to the empirical phylogeny of mammals (Fig. 1). To do this, we took the parameter estimates from model iii for all mammals, and used a forward simulation of the model to generate 100 phylogenies with large and small range categories at the tip states. The simulation was performed with the package diversitree[87]. We calculated a DR metric for every species on the simulated phylogenies, and explored the relationship between the DR and range size category using a standard linear regression and a phylogenetic linear model with Pagel lambda correlation. We then compared the relationship between DR and range sizes in the simulated phylogenies with the empirical relationship on the mammalian phylogeny (with range size discretized along the median, unlike in Fig. 1).

## Reporting summary

Further information on research design is available in the Nature Portfolio Reporting Summary linked to this article.

## Data availability

The phylogeny used for our analyses is available as a supplementary material of Upham et al. (2019)[51], and the primary range data are available from IUCN red list database[75]. Pre-processed phylogeny and range size data are in a permanent archive accompanying this paper (https://doi.org/10.5281/zenodo.8186544[93]). Source data for Figs. 1, 3 and 4 are provided with this paper.

## Code availability

A complete pipeline of data analysis is available in a permanent archive accompanying this paper (https://doi.org/10.5281/zenodo.8186544[93]).

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

## Acknowledgements
We thank L. Herrera-Alsina, A. Macháč, M. Smyčková, G.I. Ridder, W. Thuiller and S. Lavergne for helpful comments and insights at different stages of the project. The research was funded by the Czech Science Foundation (GAČR 20-29554X, D.S.).

## Author contributions
J.S. and D.S. conceived the study. J.S. designed and performed the evolutionary analyses. A.T. compiled and processed the geographical data. J.S and D.S. wrote the first draft. All authors contributed to the final version of the manuscript.

## Competing interests
The authors declare no competing interests.
