## [Peer Review File · Nature Communications]

The relationship between geographic range size and rates of species diversificationREVIEWER COMMENTS

Reviewer #1 (Remarks to the Author):

I enjoyed reading this - it is an interesting analysis and uses a neat approach to make some strong points about diversification rates and the evolution of range size in diversification models. I do think the story of the research is rather oddly presented and I have a few methodological concerns that it would be good to see addressed.

On that first point - this is your paper and you get to present it how you want! However, my experience reading it was that through the abstract, introduction and methods, I kept stopping and thinking about how the inheritance of range size could underpin contradictory patterns, in particular the fragmentation of large ranges into multiple inherited small ranges. The abstract doesn't really talk about the inheritance of range size and closes with the statement that diversification rates "are strongly influenced by idiosyncratic and spatially localized events". That conclusion isn't even remotely surprising to me - I would expect bio-geographic patterns of diversification to be strongly influenced in just this way.

The story of the research only really comes together in the results and discussion. Here, you talk very explicitly about the inheritance of range size in speciation events, and how the temporal signals of speciation used in standard models are not well-suited to modelling range size. Your models support the expectation of higher diversification at large range size, and the fact that the models "explicitly account for range size changes during speciation" - and hence can start to better explain why some observations of fast diversification at small range size may be misleading - is a big deal that is rather undersold! I would personally use this argument right from the abstract. That is a more methodological spin on the research - but the abstract at the moment doesn't really get at the novelty of what you've done and seems rather unsurprising in what it states.

The methodological concerns are:

1. Why use both OLS and PGLS?

The simulations presented in Figure 4 suggest a strong phylogenetic signal as the slopes are very different between OLS and PGLS. The lambda values are not reported: are ML estimates of lambda used in each case or a fixed lambda value? PGLS seems at face value to be the more appropriate choice here - the data is phylogenetically structured - and probably more conservative, so why present OLS?

2. The lack of a large parent to two small daughters rate

This seems like an issue to me. Peripatry typically has one small ranged daughter, but vicariance could easily give rise to 2 small ranged daughters for large ranged parents that are not much bigger than the median size. I'm not sure I follow the justification in the following sentence either - the right skew means that there are many species close to the median and fewer species with ultra-large ranges that would most likely give rise to large ranged daughters? The scenario of a large-ranged species fragmenting to give a multiple small species is constrained by this restriction, only 1 daughter at a time can become small-ranged.

If the parameter is completely unidentifiable, that may be outside your control, but I'm not currently convinced that this is an unlikely event. I guess you could simulate vicariance events on known ranges across the range size distribution and see what the frequency of different outcomes are - but I suspect L -> SS is common.

3. The whole large/small ranged discretization.

You can't avoid this and use the method but it initially seems a wildly simplistic thing to do. I don't think it is but I think it might be good to make the sensitivity analysis more central rather

than in the SM. That boundary is the critical choice in the analysis.

For example, SF1 and SF2 are about comparing estimates across different range size class definitions and this is much harder with two separate panels.

Rather than duplicating Fig 3 for each range size boundary, you could create one plot with groups as rows and columns of all three alternative range class size boundaries. They are then much easier to contrast and that joint plot could replace Figure 3.

Is it possible to use more than 2 range size classes, to try and get a little closer to the distribution? I suspect it is theoretically possible (not just binary classes in SSE?) but is not likely to be estimable. It might be good to explain that briefly.

Although this is partly covered by the use of three alternative range size divisions, I'm curious about the differences in median range sizes within your sub-clades, given the link between body size and range size. Artiodactyla are nearly all large-bodied (and large-ranged?) and bats are small-bodied (and small-ranged?). The numbers in Figure 3 actually suggest I'm wrong and that isn't too imbalanced across clades, but showing whether diversification changes for large or small range-sized species where the range size is defined relative to the pool of species being studied seems attractive.

More scattered points:

Abstract

L13 "often assumed to determine diversification rate."

Influence not determine? More than one factor!

L17 and Paragraph L42 "indicates high diversification of small-ranged species"

Or possibly a large ranged species has split into many small ranged species. One of the features of range dissection is you end up with two smaller ranged species. It could still be that large range has promoted high diversification, but that the large range no longer exists.

This is addressed in L60 and below, although I am not sure that the pattern of range fragmentation is inherently associated with short branches. It would be if ranges tend to become larger through time, because then rapid fragmentation is required to maintain small size, but that seems like an assumption about range size evolution.

Indeed, in your discussion, L165 and below explicitly explain how this pattern can arise from division of large-ranged ancestors. This is really interesting and feels like it should form the core of the way the research is presented.

L57 "such as DR metric, BAMM estimates, MoM or phylogenetic endemism"

Lots of jargon here - not really explained

Results

L110: I'd like to see the rates in the main MS here rather than in SF3 - as with the estimates of a linear model, they are a key part of the findings. The magnitudes of the μ_L estimates for primates and Eulipotyphla are disconcerting and suggest a problem with model fitting. Can the mapping of the two regimes on the phylogeny be recovered to identify which groups and species are associated with each of the two regimes?

Discussion

For me, the story of the research only really comes together in the discussion. The key points about what are appropriate range size diversification and the biogeographic processes of speciation are presented clearly here and the various circumstances under which we expect (and widely observe) departures are nicely exemplified.

While the examples are 'just' a list of interesting examples that are congruent with the explanation, I think it is correct that a formal analysis of the phenomenon (L226) is difficult (at best!) and they are useful discussion points even without a formal quantitative framework.

The discussion is a little repetitive - the main findings and reviews of sub-clade results are revisited in introducing successive broader points - and I think the writing in general in the discussion should be streamlined.

Methods

Data sources clear and cleanly described.

L334: I don't easily follow the description here

L378 and 393: Maybe introduce null models first - general case and then how particular states are tied to the observed range-sizes.

Cetartiodactyla seems unnecessary since you are only using terrestrial mammals. Artiodactyla would be clearer?

L418: Good to know difference for Primates but feels like supplementary material!

L435: R jargon - not well explained - but could be supplementary material.

Paragraph 438: minor grammatical issues but also hard to follow what you are doing and why. Is this a test of the power of your models to retrieve particular signals? This is clearer after reading the results but if - like me - you read intro, methods, results and discussion, this doesn't stand alone.

Figures:

Figure 2. The legend is a very long recapitulation of the methods and I don't think the actual figure adds much to that second explanation. It is hard to create a good diagram of these concepts, but I think this needs revisiting.

General writing notes

Very clearly written but with widespread minor grammatical issues (examples below), some run on sentences with multiple clauses (L32) and some overlong paragraphs (L268).

L31: remove 'already'

L32: should be "rates ... are"

L38: remove 'a'

L320 and 323: add 'the'

L381: 'Similarly as' to 'Like'

L334: colon after 'lineage can'

Reviewer #2 (Remarks to the Author):

The work by Smyčka et al. titled *Do the species with large geographic ranges diversify faster?*, analyze the relationship between geographic range and rates of diversification of mammalian species. This is an exciting topic and is widely studied and debated in the literature. Trying to establish the relationship between the geographic range and diversification rates is a complex topic, which leads me to have several reservations about the article.

First, the geographic range and the geographic limits of the species are not known. Recently, Parson et al. (2022) have shown that there is a large number of cryptic species within mammals. This is directly related to the geographic range of species. Smyčka's work does not consider that widely distributed species may be made up of multiple species from smaller geographic areas. An interesting point would be to incorporate the Linnean (taxonomic) and Wallacean (geographical) shortfalls in the analyses.

On the other hand, the results presented have already been indirectly shown by previous works. For example, Morales-Barbero et al. (2021) showed that climatically unstable regions have higher speciation rates. Morales-Barbero shows that the high latitudes have lineages with higher speciation rates. This is directly related to Rapoport's rule, where at high latitudes mammal species have larger geographic areas and it is in these regions where there are the highest rates of speciation. I consider associating the results found with Rapoport's rule and previous results in agreement with the authors would be important.

Finally, the work ignores the processes of sympatric speciation. Sympatric speciation may also be associated with changes in the geographic ranges of the species.

Parsons et al. (2022) PNAS, 2022; 119(14) e2103400119.

Morales-Barbero et al (2021) J Evol Biol. 2021;34:339–351.

Reviewer #1 (Remarks to the Author):

I enjoyed reading this - it is an interesting analysis and uses a neat approach to make some strong points about diversification rates and the evolution of range size in diversification models. I do think the story of the research is rather oddly presented and I have a few methodological concerns that it would be good to see addressed.

Thank you, we have followed your recommendations and addressed all the issues. We feel our manuscript has considerably improved due to these changes.

On that first point - this is your paper and you get to present it how you want! However, my experience reading it was that through the abstract, introduction and methods, I kept stopping and thinking about how the inheritance of range size could underpin contradictory patterns, in particular the fragmentation of large ranges into multiple inherited small ranges. The abstract doesn't really talk about the inheritance of range size and closes with the statement that diversification rates "are strongly influenced by idiosyncratic and spatially localized events". That conclusion isn't even remotely surprising to me - I would expect bio-geographic patterns of diversification to be strongly influenced in just this way. The story of the research only really comes together in the results and discussion. Here, you talk very explicitly about the inheritance of range size in speciation events, and how the temporal signals of speciation used in standard models are not well-suited to modelling range size. Your models support the expectation of higher diversification at large range size, and the fact that the models "explicitly account for range size changes during speciation" - and hence can start to better explain why some observations of fast diversification at small range size may be misleading - is a big deal that is rather undersold! I would personally use this argument right from the abstract. That is a more methodological spin on the research - but the abstract at the moment doesn't really get at the novelty of what you've done and seems rather unsurprising in what it states.

We agree that the methodological novelty of the paper was not properly advertised, this is also in line with the comments from reviewer #2. We have now added a sentence on the range size changes at speciation to the Abstract (l 15-19). At the same time, we have decided to keep the last sentence of the Abstract, because although geographic idiosyncracies are generally believed to be important, this importance is seldom shown as explicitly as in our paper. We have also rewritten the second half of Introduction (l 57-97) to better accommodate the argumentation about range size changes during and outside the speciation process, and the phylogenetic patterns it may generate.

The methodological concerns are:

1. Why use both OLS and PGLS?

The simulations presented in Figure 4 suggest a strong phylogenetic signal as the slopes are very different between OLS and PGLS. The lambda values are not reported: are ML estimates of lambda used in each case or a fixed lambda value? PGLS seems at face value to be the more appropriate choice here - the data is phylogenetically structured - and probably more conservative, so why present OLS?

The reason we used also OLS approach is that tip-level diversification metrics are often used without acknowledging their phylogenetic autocorrelation both for visual inspection (e.g. Jetz et al. 2012 doi:10.1038/nature11631, fig. 4; Igea et al. 2020 doi:10.1111/ele.13476, figure S9) and for spatial aggregation (e.g. Quintero et al. 2018 doi:10.1038/nature25794, Tietje et al. 2022 doi:10.1073/pnas.2120662119, Machac 2020 doi:10.1093/sysbio/syaa028) or statistical tests (e.g. van Els et al. 2021 doi:10.1038/s41559-021-01515-y). Moreover, we think that both approaches, not only OLS, have methodological downsides. The problem with using OLS is clear – we analyze

species-level characteristic with strong phylogenetic signal (lambdas close to 1 for all of our datasets) without acknowledging it. However, the problem with using PGLS is not less severe, since PGLS deals with the autocorrelation using trait evolution models that are unrealistic for the DR. The DR metric is not a classical trait, but a value derived from the phylogeny itself, with rather specific properties. For instance: (1) DR is only defined for terminal species, not along the branches; (2) terminal sister species have always identical values of DR; (3) clustered species have autocorrelated values, which are at the same time necessarily high; etc. Approximating the phylogenetic autocorrelation of DR using rescaled Brownian trait evolution in PGLS thus does not seem correct either. We do not advocate here the use of OLS or PGLS as a better methodological choice for analyzing the DR metric, but we still think it is good to report results of both the approaches, simply to relate our analyses to the previous literature.

In the current version of the manuscript, we added the slopes and lambda estimates to the Figure 1 as suggested. We, however, prefer to keep the OLS regression lines in the Figure 1, as the PGLS lines are generally underestimating the DR and thus counter-intuitively running below the point clouds (see Supplementary Figure 1). This is due to the nature of the DR metric, as the species with high DR are necessarily surrounded by other high-DR species and such clusters are systematically pulled up by the autocorrelation structure. We explained this phenomenon in the caption of newly created Supplementary Figure 1 containing both regression lines, but we think that putting the PGLS lines directly into the Figure 1 would create unnecessary confusion. We have, however, removed the confidence intervals of the OLS from the Figure 1, as these might have suggested that some serious statistical testing took place, while the regression lines were meant as a graphical guideline showing that the point values are declining.

2. The lack of a large parent to two small daughters rate

This seems like an issue to me. Peripatry typically has one small ranged daughter, but vicariance could easily give rise to 2 small ranged daughters for large ranged parents that are not much bigger than the median size. I'm not sure I follow the justification in the following sentence either -the right skew means that there are many species close to the median and fewer species with ultra-large ranges that would most likely give rise to large ranged daughters? The scenario of a large-ranged species fragmenting to give a multiple small species is constrained by this restriction, only 1 daughter at a time can become small-ranged. If the parameter is completely unidentifiable, that may be outside your control, but I'm not currently convinced that this is an unlikely event. I guess you could simulate vicariance events on known ranges across the range size distribution and see what the frequency of different outcomes are - but I suspect L -> SS is common.

Thank you, we agree that our original justification was not easy to follow. The right skew means that there is a relatively large number of species that have really large ranges. Therefore, there are also many species with the range size more than twice as large as the median, for which it is impossible that both vicariant daughters would get below median at the same time. We now show the rarity of “large mother to two small daughter” events by a simulation experiment, where we divide the actual ranges using different vicariance scenarios (see Supplementary Figure 5). We show that the proportion of “large mother to two small daughter” events would be around 10% of all vicariance speciation events in the most extreme case of strict division to halves, and considerably lower in the broken stick scenarios (i.e. random division into two pieces with a uniformly distributed boundary). Please note that this proportion only refers to vicariance mechanism, while the “large mother to two small daughter” situation is virtually impossible under other supposedly common speciation mechanisms, such as peripatry. We have now expanded our argumentation in the Methods section and refer here to the simulation experiment (1416-420). We also mention the simulation experiment in the Discussion, as it nicely shows that an important number of geographic

speciation events may result in daughters staying in the same range-size category as the mother (l 201-206).

3. The whole large/small ranged discretization.

You can't avoid this and use the method but it initially seems a wildly simplistic thing to do. I don't think it is but I think it might be good to make the sensitivity analysis more central rather than in the SM. That boundary is the critical choice in the analysis.

Indeed, we see the discretization as an inevitable core part of our methodological approach, and at the same time an expectable source of criticism. In the current version of manuscript we present more explicitly the effect of alternative small/large boundaries. We describe the specific changes in the comments below.

For example, SF1 and SF2 are about comparing estimates across different range size class definitions and this is much harder with two separate panels. Rather than duplicating Fig 3 for each range size boundary, you could create one plot with groups as rows and columns of all three alternative range class size boundaries. They are then much easier to contrast and that joint plot could replace Figure 3.

Thank you, we agree that our figure presentations were not optimal. In the current version, we combine the original Supplementary Figure 1 and 2 with Figure 3, and refer to these alternative boundaries more in the main text (l 124-125 and l 148-158). For easier comparison, we also put the analysis with the global median along the other sensitivity analyses presented in the supplementary figures (current Supplementary Figures 2 and 3).

Is it possible to use more than 2 range size classes, to try and get a little closer to the distribution? I suspect it is theoretically possible (not just binary classes in SSE?) but is not likely to be estimable. It might be good to explain that briefly.

When designing our analyses, we were considering using more than two size classes. However, we came across multiple problems that finally discouraged us from going in this direction. First of all, using more range size categories would mean a dramatical increase in the number of parameters, not only in range size dependent models, but also in comparable null models. This would likely be affordable for the all-mammals phylogeny, but it would prevent us from exploring the order-specific patterns and idiosyncracies as we do in the current manuscript, because the parameters would not be identifiable for the smaller phylogenetic trees. More importantly, using only two range size categories is a rather efficient way to be agnostic about the exact mechanisms of speciation. In the current model, we assume that vicariant, peripatric and sympatric speciations sometimes cause one daughter of large-ranged species to become small-ranged. In the vicariant speciation, it may also happen that both daughters of the large ranged species become small ranged, but this is rare as discussed above. However, if we had more range size categories, we would need to specifically address how often the small daughter ends up in the smallest category (peripatry), how frequent are the drops of daughter(s) by one of few categories down (vicariant), etc. This might still be feasible with proper conditioning of the model parameters and using only large phylogenies, and in fact it is a really interesting future direction towards identifying frequencies of various types of speciations on large phylogentic scales. But again, it would dramatically change the scope of the manuscript. In the current version, we mention both these motivations in the Discussion (l 198-201), and thoroughly describe them in the Methods section (l 454-486).

Although this is partly covered by the use of three alternative range size divisions, I'm curious about the differences in median range sizes within your sub-clades, given the link between body size and

range size. Artiodactyla are nearly all large-bodied (and large-ranged?) and bats are small-bodied (and small-ranged?). The numbers in Figure 3 actually suggest I'm wrong and that isn't too imbalanced across clades, but showing whether diversification changes for large or small range-sized species where the range size is defined *_relative_* to the pool of species being studied seems attractive.

This is a good point. Different orders have different dispersal capacities, with large bodied Carnivora and Artiodactyla and flying Chiroptera having on average larger ranges than the other three. However, as you mention, there does not seem to be a systematic pattern arising between these triplets of orders. We reanalysed the data with the order-specific medians as you suggest (Supplementary Figure 2), obtaining the results that are fairly similar to the ones with the global median, with the exception of generally unstable Eulipotyphla. The observed differences among the orders thus cannot be explained by the differences in dispersal capacities and their order-specific range size distribution, or at least it does not seem to be a major factor. We have added a discussion of this phenomenon to the current version of the manuscript (l 151-152, l 213-219 and l 399-403).

More scattered points:

Abstract

L13 "often assumed to determine diversification rate." Influence not determine? More than one factor!

Thank you. We have changed "determine" to "affect" which fits the sentence even better in our opinion.

L17 and Paragraph L42 "indicates high diversification of small-ranged species" Or possibly a large ranged species has split into many small ranged species. One of the features of range dissection is you end up with two smaller ranged species. It could still be that large range has promoted high diversification, but that the large range no longer exists.

We changed "indicates" to "might indicate" in the abstract, and slightly reformulated the second paragraph of the Introduction (l 15 and l 51-55). We now discuss the possibility that small-range species with high DR metric may emerge by range splitting of large-ranged mothers in the following paragraph and throughout the manuscript. But (mis)interpreting the correlation between the metrics based on branch length and range size as a signal of fast diversification of small-ranged species is so common in the literature that we still find legitimate to dedicate a whole paragraph to it. We also believe that building the contrast between theory and this data-based evidence at the beginning of the Introduction and then resolving it in the second part of Introduction improves readability of the manuscript for an average reader, but we do not mind restructuring the Introduction even more if needed.

This is addressed in L60 and below, although I am not sure that the pattern of range fragmentation is inherently associated with short branches. It would be if ranges tend to become larger through time, because then rapid fragmentation is required to maintain small size, but that seems like an assumption about range size evolution.

This statement was not meant to be inherently true, but instead related to the cited range size evolution models from Pigot et al (2010), where range fragmentation does indeed cause an association between short branches and small ranges, similar to the one observed in the empirical data. In the current version we have rewritten the whole paragraph to better describe our expectations concerning range size evolution (l 57-73).

Indeed, in your discussion, L165 and below explicitly explain how this pattern can arise from division of large-ranged ancestors. This is really interesting and feels like it should form the core of the way the research is presented.

Thank you, we now present our expectations about range size evolution already in the Introduction (l 57-97).

L57 "such as DR metric, BAMM estimates, MoM or phylogenetic endemism" Lots of jargon here - not really explained

Thank you. In the current version, we have reformulated the sentence to avoid these abbreviations (l 57-60).

Results

L110: I'd like to see the rates in the main MS here rather than in SF3 - as with the estimates of a linear model, they are a key part of the findings. The magnitudes of the muL estimates for primates and Eulipotyphla are disconcerting and suggest a problem with model fitting. Can the mapping of the two regimes on the phylogeny be recovered to identify which groups and species are associated with each of the two regimes?

We agree that the parameter estimates are important for a specialized reader, but at the same time we think that they are not easily interpretable for an average reader of a generally focused journal as is Nature Communications. For this reason we would prefer to keep these results in the Supplementary Materials, but we can easily move them into the main results, if this is an important requirement of the reviewers or the editor.

The high muL values for Primates and Eulipotyphla do not seem to result from convergence problems – these outlier values persist with changes of starting values and tolerance thresholds of the subplex optimisation. For easier interpretation, we now add a table of the probabilities of different states at the tips of the phylogenies (Supplementary Data 3). The state large2 linked with these extremely large muL estimates in Eulipotyphla and Primates is rather likely to contain no species (probabilities < 0.1 for all species). However, for some of the species, the probabilities of being in the state large2 are not plain 0, and these species are thus predicted to have low net diversification rate (this is calculated as a mean of net diversification rate in the different states weighted by the state probabilities). In fact, these species are exactly the large-ranged slow-diversifier exceptions already discussed in the main text, i.e. *Tarsius bancanus* for Primates, and multiple species of Eulipotyphla with the highest values for *Uropsilus gracilis*. The species with very long terminal branches are notoriously difficult to interpret in parametric diversification models (Daniel Rabosky, pers. comm.). Our models interpret these species as rare instances most likely resulting from standard diversification regimes (high probability of state large1), but admit a possibility that these are rare survivors of a regime with very high extinction rate (non-zero probability of state large2). Both these interpretations are in line with typical verbal explanations of the existence of species with long terminal branches. We added a brief explanation of this phenomenon into the caption of Supplementary Figure 4.

Discussion

For me, the story of the research only really comes together in the discussion. The key points about what are appropriate range size diversification and the biogeographic processes of speciation are presented clearly here and the various circumstances under which we expect (and widely observe) departures are nicely exemplified.

Thank you! We have now put some of our expectations about the biogeographic processes from the Discussion to the Introduction and the Abstract as suggested.

While the examples are 'just' a list of interesting examples that are congruent with the explanation, I think it is correct that a formal analysis of the phenomenon (L226) is difficult (at best!) and they are useful discussion points even without a formal quantitative framework.

Thank you, we also think that the examples represent an interesting (and even necessary) complement to the general picture presented in the first two paragraphs.

The discussion is a little repetitive - the main findings and reviews of sub-clade results are revisited in introducing successive broader points - and I think the writing in general in the discussion should be streamlined.

We have streamlined the discussion where possible, and in particular we have shortened the paragraph on the conservation consequences that was indeed lengthy and repetitive (l 306-326). At the same time, we believe that these broader points (causality directions, conservation consequences) are worth discussing in the separate paragraphs.

Methods

Data sources clear and cleanly described.

Thank you.

L334: I don't easily follow the description here

We have changed the "hidden Markov process" to "branching process". While the former is a mathematically correct term, it did not add any relevant information to the sentence and was unnecessarily technical (l 386).

L378 and 393: Maybe introduce null models first - general case and then how particular states are tied to the observed range-sizes.

We slightly prefer to keep this order of presentation, as model i and model iii represent our way of thinking about the range size evolution throughout the manuscript. The null models are specifically designed to contrast models i and iii (numbers of diversification regimes, etc.), rather than just reflecting general expectations. But this can be easily changed, if it turns out that this point is a major requirement of the reviewers or the editor.

Cetartiodactyla seems unnecessary since you are only using terrestrial mammals. Artiodactyla would be clearer?

Good point, thank you.

L418: Good to know difference for Primates but feels like supplementary material!

We agree that this is a very technical information. At the same time we found it important to report somewhere, and adding one technical sentence in the Methods seems to us more convenient than creating a Supplementary material containing just one sentence.

L435: R jargon - not well explained - but could be supplementary material.

We have removed the end of the sentence, it was indeed too technical (l 524-525).

Paragraph 438: minor grammatical issues but also hard to follow what you are doing and why. Is this a test of the power of your models to retrieve particular signals? This is clearer after reading the results but if - like me - you read intro, methods, results and discussion, this doesn't stand alone.

It is a test whether we can use the SSE parameters to retrieve the empirical patterns of the relationship between range size and DR shown in Figure 1. We have expanded this paragraph to be clearer (l 528-538).

Figures:

Figure 2. The legend is a very long recapitulation of the methods and I don't think the actual figure adds much to that second explanation. It is hard to create a good diagram of these concepts, but I think this needs revisiting.

We have redrawn the Figure 2 to be clearer and more visually appealing. It still contains similar information as the Methods, but we prefer to keep it this way, as we think it is necessary to explain the model structure for the readers that read the manuscript in the order Intro>Results and figures>Discussion, as it is structured in Nature Communications.

General writing notes

Very clearly written but with widespread minor grammatical issues (examples below), some run on sentences with multiple clauses (L32) and some overlong paragraphs (L268).

L31: remove 'already'

L32: should be "rates ... are"

L38: remove 'a'

L320 and 323: add 'the'

L381: 'Similarly as' to 'Like'

L334: colon after 'lineage can'

Thank you, we have implemented all these changes, rewritten most of the complicated sentences (e.g. l 32-34, l 53-55, l 124-125, l 138-140), and considerably streamlined the discussion paragraph about the conservation consequences (l 306-326). We also fixed multiple other grammatical issues and unclear formulations throughout the text.

Reviewer #2 (Remarks to the Author):

The work by Smyčka et al. titled Do the species with large geographic ranges diversify faster? , analyze the relationship between geographic range and rates of diversification of mammalian species. This is an exciting topic and is widely studied and debated in the literature.

Thank you. We agree that the relationship between range size and diversification is one of the oldest questions of biogeography. However, we are not aware of any previous study that would address this relationship with respect to the extremely specific evolutionary heritability of range size. In line also with the suggestions of reviewer #1, we have rewritten the Abstract (l 15-19) and the Introduction (l 57-97) to point out the novelty of our approach.

Trying to establish the relationship between the geographic range and diversification rates is a complex topic, which leads me to have several reservations about the article. First, the geographic range and the geographic limits of the species are not known. Recently, Parson et al. (2022) have shown that there is a large number of cryptic species within mammals. This is directly related to the geographic range of species. Smyčka's work does not consider that widely distributed species may be made up of multiple species from smaller geographic areas. An interesting point would be to incorporate the Linnean (taxonomic) and Wallacean (geographical) shortfalls in the analyses.

This is a good point. Although we use the most up-to-date dataset that is standardly used in macroecological and macroevolutionary analyses without considering the possible biases in the data (e.g. Upham et al. 2021 doi:10.1016/j.cub.2021.07.012, Greenberg et al. 2021 doi:0.1111/ele.13868), it is fair to discuss its geographic and taxonomic uncertainties.

The Wallacean shortfall likely does not have crucial impact to our analyses. This is because we work with species categorized as “large-ranged” and “small-ranged”, and although there might be some uncertainties in the knowledge of absolute range sizes, these would only rarely lead to species crossing the large/small threshold. Moreover, we show that our main results are not sensitive to the redefinition this threshold (Figure 3 and Supplementary Figure 2). We added this line of argumentation into the newly created paragraph in the Data section of Methods (l 351-358).

The Linnean shortfall might be more important here. However, it does not seem that the cryptic species would create a systematic pattern among the separately explored orders. We can expect that cryptic species are much more common in Chiroptera, Eulipotyphla and Rodentia than in Atriodyctyla, Carnivora and Primates (Parsons et al. 2022 doi:10.1073/pnas.2103400119), but these triplets of orders do not systematically differ in the patterns of diversification based on range size.

Apart from that, to address the idea of the referee to incorporate cryptic species in the analysis, we have performed a sensitivity experiment. We artificially splitted 10% or 30% of the largest species in our all mammals dataset to the pairs of cryptic species, mimicking the situation described in Parsons et al. (2022 doi:10.1073/pnas.2103400119). The analysis of these datasets suggests that the general patterns we discuss in the manuscript can be retrieved even from the data with added cryptic species (Supplementary Figure 3), but the taxonomic definitions may still influence the exceptional lineages reported in the manuscript. We newly mention this sensitivity experiment in the Results section (l 110-112), and we have expanded the Discussion paragraph dedicated to the cryptic species (l 271-280), using the argumentation from above. We have also newly created a paragraph in the Data section of Methods, describing in detail our sensitivity experiment with cryptic species and its results (l 360-369).

On the other hand, the results presented have already been indirectly shown by previous works. For example, Morales-Barbero et al. (2021) showed that climatically unstable regions have higher

speciation rates. Morales-Barbero shows that the high latitudes have lineages with higher speciation rates. This is directly related to Rapoport's rule, where at high latitudes mammal species have larger geographic areas and it is in these regions where there are the highest rates of speciation. I consider associating the results found with Rapoport's rule and previous results in agreement with the authors would be important.

It is true that both the sizes of geographic ranges and diversification rates may be influenced by the latitude or past climatic histories, and we discuss it as a possible explanation of large-ranged slow diversifying temperate species of Eulipotyphla. However, this effect is likely not crucial at the level of the whole dataset, because the relationship between range size and latitude in our dataset is rather weak (see Supplementary Figure 6), and also the evidence of the relationship between latitude and diversification in mammals is non-conclusive and dependent on used methods (e.g. negative in Rolland et al. 2014 doi:10.1371/journal.pbio.1001775 versus positive in Morales-Barbero et al. 2021 doi:10.1111/jeb.13737). At the same time, it does not seem there would be a systematic difference in our results between mostly tropical orders (Primates, Chiroptera) versus the others, which one would expect if our results were mostly driven by the latitudinal contingencies. We now explicitly put these arguments in the Discussion (l 219-225).

Finally, the work ignores the processes of sympatric speciation. Sympatric speciation may also be associated with changes in the geographic ranges of the species.

Thank you. This is not a conceptual issue, but only a misunderstanding caused by our unclear writing. Sympatric speciations are taken into account in our analyses in the same way as allopatric speciations. We assume that the large-ranged species sometimes produce small-ranged daughters at speciation, without any further assumption on whether the speciation was triggered due to geographic barriers or intrinsic population processes. We agree that the whole manuscript is rhetorically build around the geographic processes, which may give the false impression that we omit sympatric speciation. At the same time, we sometimes confusingly referred to the allopatric mechanisms in the previous version. In the current version we have added an explicit note on sympatric and allopatric speciations in the Introduction (l 39-40) and have rewritten the parts that might give a false impression that the speciations associated with the changes of range size are only allopatric (e.g. l 183-184 and l413).

Parsons et al. (2022) PNAS, 2022; 119(14) e2103400119.

Morales-Barbero et al (2021) J Evol Biol. 2021;34:339–351.

REVIEWER COMMENTS

Reviewer #1 (Remarks to the Author):

Review

I enjoyed re-reading this and found it much easier to follow the various components of the research. I have a few comments from the response letter:

* I agree that PGLS and OLS are complementary - in my review I hadn't fully thought through that PGLS was being used on range sizes here and hence the evolutionary model from PGLS is flawed.

* Thanks for adding the simulation to look at the rarity of L -> SS daughters. I think this is much clearer and more convincing now. Incidentally, from the previous version, right-skewed means that the `_tail_` of the distribution is to the right, which is one reason why I was concerned about this. The data sound like they are left-skew, which makes the relatively rarity of that transition more convincing.

* The justification and defence of using two range size classes is good - definitely worth pre-emptively tackling that possible criticism firmly!

* It is great to have a link to the GitHub code repo. Does this need to be updated to reflect the revision. Also, is it possible to add the key input data files? Obviously, GH is not suitable for large binary data files, but if the derived range sizes and phylogenies and the like could be shared that would be important.

Figures

Fig 1.

I think the PGLS lines should be here. I can see why you omit them: their location is unintuitive, but it doesn't feel right to include PGLS in the main MS and then not present the lines. The comparable models are presented in Figure 4, for example. I guess an alternative here is to move all of the PGLS models into supplementary and say PGLS is not well suited to non-Brownian/OU range size data, but SM shows that PGLS results are at least congruent.

Fig 2.

This is clearer, thanks. I do have a couple of questions from this figure:

* It raises some questions about the null models. There are fewer states in the null models than the alternative models. I can't see how having three hidden regimes for model iv would be more interpretable, but does it matter that the alternative models have more parameters than the null models?

* Throughout the MS, I do find the i, ii, iii, ... labels for the models simply visually harder to tell apart than a simple A, B, C but that is a minor whinge.

* The connection between the vertical scale of the phylogeny and the horizontal layout on either side of the root bifurcation gives the visual impression the vertical layout of the model schema maps onto the states of the example tree, which they don't really do. It's a complex model setup and hard to capture in a figure - this doesn't quite work for me but I don't have a great suggestions. I think if I was doing this I'd have four phylogenies and annotate transitions and hidden regimes on the tips, but that would be a lot bulkier.

Figure 3.

Very much clearer, thanks! One minor thing is that there is a lot of vertical white space, making detail harder to see, driven by the need to accommodate that long median threshold tail for primates. I don't generally like truncating axes, but as in your truncation of Supp Fig 4, sometimes it is better to show all the rest of the data more clearly. It is also hard to see the violin lines against the deep blue - maybe use a paler shade?

Minor comments

The writing is generally really clear. Having said that, there are a few areas, particularly in the discussion, where there are long and rather discursive paragraphs. Writing style is a personal thing, but this isn't as snappy as it could be, and I definitely felt my eyes start to skate over some of the paragraphs. Some examples:

L249 Sentence a bit unclear:

Fast-radiating and small-ranged bats and ungulates are associated with geographic domains containing barriers that stimulate speciation while preventing species from expanding their ranges, such as archipelagos and mountains.

L260 A long paragraph - some repetition from earlier sections, could be terser.

L267 "The reasons why some species maintain a large range that did not get fragmented may be fairly variable." Could be deleted entirely - you go on to present some reasons.

L271 "It is also possible that ..."

"However, up to 30% of mammalian species may be cryptic 68, notably in the Chiroptera, Eulipotyphla and Rodentia, so we assess whether our results are robust to the addition of up to 30% of cryptic species."

"We have assessed that the general results presented in the study are robust to" -> "Our results are robust to"

"However, it is important to point out that the" -> as opposed to all the trivial stuff you are pointing out? Just not needed!

Grammar

There are some really minor grammatical issues, notably the word "the" is often used unidiomatically.

L22 "overdriven" -> "override"?

Unecessary 'the':

L31 "the daughter species"

L45 "the areas of particular interest"

L52 "the range size"

L58 "the phenomenological"

L61 "The common speciation"

L68 "the process-based models"

L368 "the imperfect"

L76 "causing that many newly emerged species are small-ranged"

-> "causing many newly emerged species to be small-ranged"

L76 "unlikely" -> "less likely"?

L89 "Equivalent approach" -> "Equivalent approaches"

L126 "resulted into" -> "resulted in"

L357 median -> the median

L365 splitted -> split

L455 "getting thus" -> providing?

Reviewer #2 (Remarks to the Author):

Firstly, I would like to thank the authors for their efforts in addressing all the questions and criticisms from the reviewers. I believe that this new version of the manuscript is more theoretically and methodologically sound than the previous one. However, I have one concern based on the authors' responses. The authors argue that "the evidence of the relationship between latitude and diversification in mammals is non-conclusive and dependent on used methods (e.g. negative in Rolland et al. 2014 doi:10.1371/journal.pbio.1001775 versus positive in Morales-Barbero et al. 2021 doi:10.1111/jeb.13737)." I wonder how we can be sure that their results are not influenced by the methods? Is it possible that working with continuous range size and using other methods for estimating the rate of diversification (e.g. BAMM) might yield conflicting results?

Finally, I would like to congratulate the authors on their work. I believe that it will be of great importance for future studies.

Reviewer #1 (Remarks to the Author):

Review

I enjoyed re-reading this and found it much easier to follow the various components of the research.

Thank you, this is really encouraging!

I have a few comments from the response letter:

* I agree that PGLS and OLS are complementary - in my review I hadn't fully thought through that PGLS was being used on range sizes here and hence the evolutionary model from PGLS is flawed.

Yes, the DR metric has really counter-intuitive properties when used as a response variable in the regression.

* Thanks for adding the simulation to look at the rarity of L -> SS daughters. I think this is much clearer and more convincing now. Incidentally, from the previous version, right-skewed means that the _tail_ of the distribution is to the right, which is one reason why I was concerned about this. The data sound like they are left-skew, which makes the relatively rarity of that transition more convincing.

Thank you for your previous suggestion, our argumentation is much more firm when supported by the simulation. However, the range site distribution is really right-skewed as we previously commented, and it is the large-ranged mothers in the heavy right tail that have no way to split into two small-ranged daughters.

* The justification and defence of using two range size classes is good – definitely worth pre-emptively tackling that possible criticism firmly!

We know that this has been a critical part of our approach. Thank you for giving us the opportunity to refine our arguments in this respect.

* It is great to have a link to the GitHub code repo. Does this need to be updated to reflect the revision. Also, is it possible to add the key input data files? Obviously, GH is not suitable for large binary data files, but if the derived range sizes and phylogenies and the like could be shared that would be important.

We have updated the GitHub repository and added the most recent supplementary analyses there. The phylogeny and range size data were already in the previous version of the repository (vertlife_mammaltree.txt, cleanmainlanddf.RData), and now we have added the range size data in the human-readable format (cleanmainlanddf.csv) and refer to them in the Data availability statement. If the paper gets accepted, we will produce a clean and final version of the repository and will turn it into a permanent Zenodo archive, according to the guidelines of Nature Communications.

Figures

Fig 1.

I think the PGLS lines should be here. I can see why you omit them: their location is unintuitive, but it doesn't feel right to include PGLS in the main MS and then not present the lines. The

comparable models _are_ presented in Figure 4, for example. I guess an alternative here is to move all of the PGLS models into supplementary and say PGLS is not well suited to non-Brownian/OU range size data, but SM shows that PGLS results are at least congruent.

Thank you, we have now added the PGLS lines into the Figure 1, together with an explanation why they do not necessarily go through the point clouds. To reduce the complexity of the figure, we now only include those regression lines whose slopes were significantly different from zero. All regression lines and their bootstrap envelopes are still present in the Supplementary Figure 1.

Fig 2.

This is clearer, thanks. I do have a couple of questions from this figure:

* It raises some questions about the null models. There are fewer states in the null models than the alternative models. I can't see how having three hidden regimes for model iv would be more interpretable, but does it matter that the alternative models have more parameters than the null models?

This is just a misunderstanding, we possibly did not describe the parameter space properly. In model (i) there are two states, large and small; in model (ii) there are 4 states, large1, large2 small1 and small2; the same applies to model (iii); and model (iv) contains 8 states. However, more important for the model explanatory power is the number of parameters. In terms of free parameters, model (ii) has 6 parameters compared to model (i) with 5 parameters; and model (iii) has 12 parameters, while model (iv) has 20 parameters (see Methods). As you point out, models (i) and (iii) are richer in the speciation parameters depicted in Fig. 2, whereas models (ii) and (iv) are richer in the state-transition parameters that are not depicted there. We have added an explanatory note on the total parameter counts in the Figure 2 caption and in the Methods. In fact, we were also exploring models with more hidden states than 4 at the beginning of the project, but we have decided not to use them, because they appeared over-parameterized for the smaller mammalian orders.

* Throughout the MS, I do find the i, ii, iii, ... labels for the models simply visually harder to tell apart than a simple A, B, C but that is a minor whinge.

We would prefer to keep the notation i, ii, iii, iv. It corresponds to the increasing complexity of the models, and at the same time it does not interfere with a and b notation of the Figure 4.

* The connection between the vertical scale of the phylogeny and the horizontal layout on either side of the root bifurcation gives the visual impression the vertical layout of the model schema maps onto the states of the example tree, which they don't really do. It's a complex model setup and hard to capture in a figure - this doesn't quite work for me but I don't have a great suggestions. I think if I was doing this I'd have four phylogenies and annotate transitions and hidden regimes on the tips, but that would be a lot bulkier.

The model schemas and their hidden regimes were indeed meant to map on the example tree. We now use a different example tree that is hopefully less confusing, and we also point this out in the caption.

Figure 3.

Very much clearer, thanks! One minor thing is that there is a lot of vertical white space, making detail harder to see, driven by the need to accommodate that long median threshold tail for primates. I

don't generally like truncating axes, but as in your truncation of Supp Fig 4, sometimes it is better to show all the rest of the data more clearly. It is also hard to see the violin lines against the deep blue - maybe use a paler shade?

Thank you for suggestions. We have redrawn the figure. We tried to use the truncated axes in the figure, but they were adding even more complexity to the already complicated figure, so we prefer to leave them as before, even though we reduced the vertical white space. We now use a paler blue shade in the figure as suggested.

Minor comments

The writing is generally really clear. Having said that, there are a few areas, particularly in the discussion, where there are long and rather discursive paragraphs. Writing style is a personal thing, but this isn't as snappy as it could be, and I definitely felt my eyes start to skate over some of the paragraphs. Some examples:

Thank you. We have addressed the points raised below and further improved the text flow in the Discussion.

L249 Sentence a bit unclear: Fast-radiating and small-ranged bats and ungulates are associated with geographic domains containing barriers that stimulate speciation while preventing species from expanding their ranges, such as archipelagos and mountains.

Thank you, we have modified the formulation.

L260 A long paragraph - some repetition from earlier sections, could be terser.

We have shortened the paragraph where possible, including your suggestions below.

L267 "The reasons why some species maintain a large range that did not get fragmented may be fairly variable." Could be deleted entirely - you go on to present some reasons.

OK, we have removed it.

L271 "It is also possible that ..." → "However, up to 30% of mammalian species may be cryptic 68, notably in the Chiroptera, Eulipotyphla and Rodentia, so we assess whether our results are robust to the addition of up to 30% of cryptic species."

We have reformulated the sentence in this direction.

"We have assessed that the general results presented in the study are robust to" -> "Our results are robust to"

OK, done.

"However, it is important to point out that the" -> as opposed to all the trivial stuff you are pointing out? Just not needed!

It was indeed redundant, thank you.

Grammar

There are some really minor grammatical issues, notably the word "the" is often used unidiomatically.

Thank you. We have fixed the issues listed below, and asked our American colleague to proofread the manuscript.

L22 "overdriven" -> "override"?

Unecessary 'the':

L31 "the daughter species"

L45 "the areas of particular interest"

L52 "the range size"

L58 "the phenomenological"

L61 "The common speciation"

L68 "the process-based models"

L368 "the imperfect"

L76 "causing that many newly emerged species are small-ranged"-> "causing many newly emerged species to be small-ranged"

L76 "unlikely" -> "less likely"?

L89 "Equivalent approach" -> "Equivalent approaches"

L126 "resulted into" -> "resulted in"

L357 median -> the median

L365 splitted -> split

L455 "getting thus" -> providing?

Reviewer #2 (Remarks to the Author):

Firstly, I would like to thank the authors for their efforts in addressing all the questions and criticisms from the reviewers. I believe that this new version of the manuscript is more theoretically and methodologically sound than the previous one.

Thank you. The comments in the previous versions were really helpful and have improved the manuscript.

However, I have one concern based on the authors' responses. The authors argue that "the evidence of the relationship between latitude and diversification in mammals is non-conclusive and dependent on used methods (e.g. negative in Rolland et al. 2014 doi:10.1371/journal.pbio.1001775 versus positive in Morales-Barbero et al. 2021 doi:10.1111/jeb.13737)." I wonder how we can be sure that their results are not influenced by the methods? Is it possible that working with continuous range size and using other methods for estimating the rate of diversification (e.g. BAMM) might yield conflicting results?

We have shown that the state-dependent methods and DR metric indeed yield different results, and we have used simulations (Fig. 4) to show that the DR results are biased, in contrast to the state-dependent models. We originally did not include BAMM analysis in our manuscript due to high computational demands, and also because we expected the results to be very similar to DR metric. BAMM and DR are known to be highly correlated (Title and Rabosky doi.org:10.1111/2041-210X.13153), and also the questionable strategy of post processing diversification rate with regression is the same when using BAMM and DR metric. We now include the analysis based on BAMM, showing generally the same direction of results as the DR metric, although weaker and less significant (Supplementary Figure 2). This is likely caused by the fact that BAMM is fairly conservative in attributing rate changes to individual nodes, and many smaller mammalian order were only attributed one or a few rate changes (especially Carnivora, Artiodactyla and Primates). Anyway, these additional analyses match our original conclusion that the methods based on tip-rate postprocessing (DR, BAMM) tend to show negative relationship between diversification rate and range size, but this relationship is an artefact caused by improper treatment of range size heritability. We also add a remark on BAMM analysis in the Introduction, Discussion and the Methods sections.

Finally, I would like to congratulate the authors on their work. I believe that it will be of great importance for future studies.

Thank you very much for your positive assessment.